# Type 2 Diabetes Mellitus and *Demodex folliculorum* Infestation: A Cross-Sectional Study in Peruvian Patients

**DOI:** 10.3390/ijerph192013582

**Published:** 2022-10-20

**Authors:** Guido Bendezu-Quispe, José Rojas-Zevallos, Jaime Rosales-Rimache

**Affiliations:** 1Escuela de Medicina, Universidad César Vallejo, Trujillo 13001, Peru; 2Hospital de Apoyo de Palpa, Ica 11330, Peru; 3Facultad de Medicina, Universidad Privada Norbert Wiener, Lima 15046, Peru

**Keywords:** *Demodex folliculorum*, demodicosis, type 2 diabetes mellitus, cyanoacrylate

## Abstract

Background. *Demodex folliculorum* is a mite that causes human demodicosis, which is characterized by producing lesions on the face, and its infestation could be associated with factors that alter the immune response, including Type 2 Diabetes (T2D). This study aimed to evaluate the relationship between T2D and *D. folliculorum* infestation in patients attended at a hospital in Peru in 2021. Methods. A cross-sectional study enrolling patients who were classified according to the presence (n = 100) and absence (n = 100) of T2D was conducted. Information was obtained on age, sex, domestic animal husbandry, and symptoms and signs associated with demodicosis. Demodicosis was determined to be present when counts of *D. folliculorum* greater than five mites per cm^2^ were found in superficial facial skin biopsies. Results. A frequency of demodicosis in 76 and 35 patients with and without T2D, respectively, was observed. In the multivariate analysis adjusted for confounders age, sex, and domestic animal husbandry, T2D was found to be associated with *D. folliculorum* infestation (PR: 2.13; 95% CI95: 1.42–3.19). Conclusions. T2D was associated with *D. folliculorum* infestation. In this sense, it is important to strengthen surveillance to identify of *D. folliculorum* infestation in people with T2D.

## 1. Background

*Demodex* sp. involves ectoparasites that produce human demodicosis, which is characterized by skin lesions at the level of the pilosebaceous units on the face and head [1] when a high density of *Demodex* mites is presented in an individual. Many conditions related to an imbalance in the immune system could increase the reproduction of these mites and the presence of clinical manifestations [2,3]. *D. folliculorum* is the most common *Demodex* sp. affecting humans, and its distribution is universal; however, it has been reported mostly among adolescents and children [4]. *D. brevis* is the other *Demodex* sp. affecting humans [2]. These ectoparasites reside in the ocular and skin pilosebaceous complexes, occupying different topographical sites (*D. folliculorum* [0.3–0.4 mm] focused on the hair follicle; *D. brevis* [0.2–0.3 mm] on the sebaceous gland) [2,3]. Its presence is believed to be associated with exposure to animals such as cats, dogs, and rodents, among others; and, to the lack of personal hygiene and person-to-person contact or fomites (towels, pillows, clothes, etc.) or objects that are contaminated with any stage of the mite (eggs, larvae, protonymphs, and adults) [2], however, with inconclusive results. Likewise, a higher risk of infestation has been found in people with a greater number of sebaceous glands and variations in their immune status [5].

Studies carried out in the Latin American and Caribbean region refer to a prevalence of demodicosis of 72%, mainly by *D. folliculorum* [6], and its association with rosacea (OR = 9.3, *p* < 0.001) [7] and severe acne vulgaris (OR = 4.2, *p* < 0.01) [8]. *Demodex* species infestation is described in the literature as associated with many ocular pathologies, including blepharitis, eyelash loss and abnormal eyelash alignment, dry eye, meibomitis, chalazion, and pterygium [2,9]. Furthermore, high frequencies of demodicosis have been found in people with altered immune statuses, such as chronic kidney failure, cancer (urological, basal cell carcinoma, and epidermal neoplasms on the face), childhood malnutrition, rheumatoid arthritis, and type 2 diabetes (T2D), especially when glycemic control is poor [10]. Specific phenotypes of the HLA system have been found more susceptible to demodicosis infestation, so the immune response is key in understanding its pathophysiology [11]. Furthermore, it is described that *Demodex* infestation increases with age, being more prevalent in the general population over 60 years [12]. In addition, Demodex profiles could be different between young and older people [12,13].

T2D is considered a global public health problem [14], and its approximate prevalence is 2.5% [15]. By 2019, diabetes was the ninth leading cause of death worldwide, responsible for 1.5 million deaths [16]. Around 48% of deaths due to diabetes occur in people under 70 years [16]. In low- and middle-income countries, premature mortality due to diabetes has increased in the last decades [16]. Several studies have shown the possible relationship between T2D and *D. folliculorum* infestation with a marked increase in mite density in patients with hyperglycemia [17,18], especially at the level of the face in people with concomitant conditions such as allergic rhinitis [19] and ocular surface in people with dry eye disease [20]. T2D produces a low and chronic inflammatory response that generates immune dysfunction during hyperglycemia, increasing susceptibility to opportunistic infections [21].

In Peru, two new cases of T2D are registered for every 100 people per year [22], with a prevalence of 4.5% [23]. In this country, by 2018, T2D represented the sixth leading cause of mortality in the general population [23]. The increase in overweight and obesity in Peruvian people could explain the increased proportion of the Peruvian population affected by T2D [23]. In Peru, public health policies aimed at the prevention, control, and treatment of T2D are limited [24], and access to quality sanitary conditions, such as safe water, is not available to the entire population, influencing hygiene measures [25]. Considering that T2D is a prevalent disease in Peru and could increase the risk of *D. folliculorum* infestation, this study aimed to evaluate the relationship between T2D and *D. folliculorum* infestation. This information is critical to identify a risk group for demodicosis and improve prevention and control strategies for this disease, as well as the timely identification of cases through sensitive and specific tests.

## 2. Methods

### 2.1. Study Area and Participants

Between March and September 2021, for convenience, 100 patients with a confirmatory diagnosis of T2D established between 2019 and 2020 using the American Diabetes Association criteria (hemoglobin A1c [HbA1C] of greater than or equal to 6.5% or fasting blood glucose of greater than or equal to 126 mg/dL or two-hour blood glucose of greater than or equal to 200 mg/dL) [26] registered in the non-communicable disease program of the Hospital of Palpa were enrolled.

As a comparison group, for convenience, 100 patients who underwent laboratory control under 12 h fasting conditions and had results in the normal range (glycemia: 70–110 mg/dL, and HbA1c < 6.0%) and had no clinical manifestations associated with T2D, were enrolled. Palpa is a province of the Ica region (located at 92.7 km from Ica city, the capital of this administrative region) that does not have access to the sea, and its main economic activity is agriculture. The Palpan population lives mainly in rural conditions.

Patients under 18 years of age and patients with skin lesions on the face associated with herpes, rosacea, dermatitis, and systemic lupus erythematosus were excluded. Likewise, pregnant women and patients who had been receiving topical treatment with acaricides in the last 30 days were also not considered in the study.

### 2.2. Techniques and Procedures

A data collection sheet was used to register information on characteristics of the participants divided into two groups (patients with T2D and without T2D): age (categorized into age groups according to the WHO classification), sex, raising domestic animals, and the presence of symptoms or signs associated with demodicosis (at least one of the following: burning, erythema, rashes and itching in the face, eyelash loss, blepharitis, and other identified symptoms).

A superficial biopsy was obtained for the identification of *D. folliculorum* [27]. The standardized superficial biopsy technique with cyanoacrylate was used (a non-invasive diagnostic method in which the sample is taken from the same area in duplicate in an area of 1 cm^2^). This technique reaches a specificity and sensitivity of 95.5% and 98.7% [28]. It was executed as proposed by Forton F. et al. [29]. A drop of cyanoacrylate on a slide that adheres to the patient’s face was applied, covering an approximate surface of 1 cm^2^, which was pressed for a minute, then it was detached from the skin, and a drop of oil was added. After an immersion of the sample obtained in the slide, a coverslip was placed, and it was observed at different magnifications (20× and 40×) under the microscope (Figure 1). The number of mites per cm^2^ was reported. A count greater than 5 mites per cm^2^ was considered positive. The processing of the samples was carried out immediately because the movement of the parasite decreased, and they disintegrated over time, so a maximum time of 4 h was established for their analysis. The reading of the slides was performed single-blind by two microscopists with experience in recognizing ectoparasites and who were unaware if the slides came from the group of patients with and without T2D.

### 2.3. Statistical Analysis

The power obtained by 200 patients was 92.1%, calculated in the software PASS (Power Analysis and Sample Size) version 11, using a logistic regression model at a significance level of 0.05 and a proportion of demodicosis in people with controlled and uncontrolled T2D of 0.111 and 0.333 as reported by Gökçe et al. [17] generating an odds ratio of 4.0. Multiple regression adjustment was performed with an R-squared of 0.7.

The results were presented using descriptive statistics by the study group. The *D. folliculorum* count was compared for each independent variable in the two study groups using Pearson’s chi-square test. In a complementary manner, the counts of *D. folliculorum* were compared using the non-parametric Mann–Whitney test. A *p*-value < 0.05 was considered a significant difference. Due to the high proportion of D. folliculorum infestation found in both groups, the relationship between T2D and *D. folliculorum* infestation was evaluated in a generalized linear model with log link and Poisson family to obtain the prevalence ratio (PR) and adjusted prevalence ratio (aPR). The corresponding 95% confidence intervals (95%CI) for PR and aPR were estimated. The selection of variables for the model was based on epidemiological criteria. No missing data were observed in the study variables. Calculations were performed using the software Stata v.17 (Stata Corp College Station, TX, USA).

### 2.4. Ethical Aspects

The study was approved on 30 September 2020, by the Palpa Hospital Research Office, Regional Health Directorate from Ica, Peru (Code N°: 688-2021-GORE/DIRESA-HAP). The participation of the patients in the study was subject to the granting of their informed consent, where they were informed of the objective, benefits, risks, procedures, and techniques to be applied to the participants, and it was conducted following Helsinki declaration guidelines. The results were given to each participant within 24 h after sampling, and people with demodicosis received medical attention and acaricide treatment with recommendations to avoid future reinfestations by *D. folliculorum*.

## 3. Results

In total, 100 people with T2D and 100 people without T2D were evaluated. Age means (T2D: 57.5 ± 11.2 versus without T2D: 55.0 ± 11.4; the age range was 25–95 years), and sex distribution (male: 41[T2D] versus 42[without T2D]) were similar in both groups. A high frequency of people raising domestic animals was found in both groups (T2DM: 75 versus without T2D: 81). The presence of symptoms associated with *D. folliculorum* infestation was low in both groups (four cases in people with T2D and one case in people without T2D) (Table 1). The only symptom reported by the participants was an itchy face. Laboratory analysis showed that *D. folliculorum* infestation had a frequency of 76% and 35% in patients with and without T2D, respectively. No *D. brevis* infestation was found in the evaluated patients.

There was no difference in the *D. folliculorum* infestation presence according to age group (<60 years or more, *p* = 0.485), sex (*p* = 0.758), or domestic animal husbandry (*p* = 0.376). A difference in *D. folliculorum* infestation was found according to the presence of symptoms of the disease (*p* < 0.05). The median *D. folliculorum* count in patients with and without symptoms was 69 [IQR: 63–74] and 7 [IQR: 2–14], respectively. Regarding T2D, a significant difference (*p* < 0.001) in the presence of demodicosis was found (patients with and without T2D had median *D. folliculorum* counts of 15 [IQR: 6.5–27.5] and 3 [IQR: 1–8], *p* < 0.001 [Mann–Whitney Test]) (Table 2).

The crude analysis found an association between T2D and *D. folliculorum* infestation (PR: 2.17; 95%CI: 1.46–3.24). On the multivariate analysis, a significant association was observed between T2D and *D. folliculorum* infestation (*p* < 0.001), those patients with T2D being 2.13 times likely to suffer from *D. folliculorum* infestation compared to those without T2D (aPR: 2.13; 95% CI: 1.42–3.19, adjusted by sex, age, and domestic animal husbandry).

## 4. Discussion

The aim of this study was to evaluate the association between T2D and *D. folliculorum* infestation. Our results show a serious problem of infestation by *D. folliculorum*, in people suffering from T2D; and even in those without this disease in the province of Palpa, in the region of Ica, Peru. Furthermore, it was found that three out of four people with T2D have the presence of at least five adults of *D. folliculorum* per cm^2^ when sampling with the cyanoacrylate biopsy technique. In fact, T2D was found to be associated with *D. folliculorum* infestation since those affected have more than twice the risk of contracting *D. folliculorum* infestation compared to people without T2D.

There was no difference in the presence of *D. folliculorum* infestation according to age; even though other studies have found that its frequency is higher among the elderly compared to the young [11,30,31], who would present a lower prevalence of this health problem [12]. The risk of *D. folliculorum* infestation or demodicosis with an increased age could be explained by changes in sebaceous glands and sebum production throughout life, including hyperplasia of sebaceous glands in the face at an advanced age; except for men whose important changes occur after 80 years [32]. The presence of *D. folliculorum* infestation or demodicosis should be evaluated, especially in older adult patients, in whom cataract surgery is more frequent, due to the complications that demodicosis can generate in post-surgery patients [33]. Hence, although no difference was found in the participants analyzed in the presence of *D. folliculorum* infestation according to age, clinicians evaluating patients with non-specific discomfort for visual problems should take into consideration the possible infestation by *D. folliculorum* infestation or demodicosis, especially in older patients, for whom the literature indicates a higher probability of infestation by those parasites.

According to sex, there was no difference in the presence of *D. folliculorum* infestation. Information regarding the presence of *D. folliculorum* infestation according to sex is scarce. A study was conducted in China where sex was not associated with demodicosis infestation [31]. The same is true for conducted in an Argentinian population [2]. However, another study conducted found differences in the colonization of *Demodex* sp. species according to sex, where males were more colonized than females (23% versus 13%) and presented more colonization by *D. brevis* compared to females’ counterparts. Regarding domestic animal husbandry, it is presumed that dogs, cats, and other domestic animals can be carriers of *D. folliculorum,* so it is reasonable to think that it could be associated with human demodicosis. However, in the study sample, there was no difference in the presence of *D. folliculorum* infestation according to domestic animal husbandry. This finding could be supported by the fact that human and canine demodicosis is caused by different Demodex species even though the immune responses and evasion mechanisms are similar [34]. Hence, in the study participants, the transmission of Demodex mites could be produced in human-to-human interaction and not related to domestic animal husbandry.

Regarding the symptoms associated with *D. folliculorum* infestation, an itchy face was the only symptom reported (five cases). It is important to mention that most people with *D. folliculorum* are carriers, not presenting symptoms attributable to this infestation. Although *D. folliculorum* infestation or demodicosis symptoms are non-specific, some occur more frequently, such as itching, scaling, and folliculitis [35]. In fact, the latter, due to the blockage of the hair follicles, can generate allergic reactions and local inflammation and be associated with rosacea, nonspecific facial dermatitis, steroid rosacea, androgenetic alopecia, madarosis, lupus miliaris, and folliculitis dissecans, among others [36]. It is described that a transition from a non-symptomatic *Demodex* infestation to a clinically dermatological disease could be produced by the developing of primary or secondary immunodepression, being that humans and animals with immunosuppression have an increased probability of presenting *Demodex* infestations [37,38]. Although only a few of the patients included in the study presented symptoms, the clinicians should consider *D. folliculorum* infestation or demodicosis as a possible disease in patients reporting an itchy face or other dermatological or ocular pathology described in the literature related to *D. folliculorum* infestation or demodicosis.

Our results show a high frequency of *D. folliculorum* infestation in people with T2D compared to that found in people without T2D. These figures exceed those reported in other diabetic populations that have had frequencies that oscillate between approximately 24 and 34% [17,20]; although Yamashita L. et al. reported a prevalence of demodicosis of 54.8% in patients with T2D (control group: 38.1%, *p* < 0.05). The high frequency of demodicosis may be because generated by T2D on the microvasculature and altered immune response [21]. Several studies have indicated that the composition of the sebum produced by the sebaceous glands changes in people with T2D [39], consequently, also the residual components of the epidermis [40], reducing the protective function of the skin. Another critical aspect of T2D is glycemic control. There is evidence that indicates that demodicosis frequencies of 17.6% and 82.4% have been found in people with T2D who had adequate and poor glycemic control, respectively [17].

This study found a significant association between T2D and *D. folliculorum* infestation, and it is observed that people with T2D have more than twice the risk of having *D. folliculorum* infestation compared to those who do not have T2D after adjusting this association by sex, age, and domestic animal husbandry. Although this infestation does not represent a significant risk for the development of major health problems or mortality, in the case of people with T2D, whose inflammatory response is altered [21], the complications related to the progression of T2D could increase the risk of *D. folliculorum* infestation. Furthermore, it is widely known that in patients with T2D, blood glucose regulation is critical to prevent diabetes complications. Regarding demodicosis, it is also described that an unregulated blood glucose level increases the susceptibility to *D. folliculorum* infestation [17]. Hence, the screening for *D. folliculorum* infestation or demodicosis in patients with T2D could be relevant to preventing more complications, including the development of secondary bacterial or fungal infections, blepharitis, and rosacea, among others [41,42].

The study’s limitations are attributed to the cross-sectional design, whose lack of temporality did not allow for the evaluation of causality between *D. folliculorum* infestation and T2D. Furthermore, there are groups at risk for *D. folliculorum* infestation or demodicosis which were not included in this research, such as pregnant women, in whom *D. folliculorum* infestation or demodicosis presence could be more prevalent, especially in those with poor glycemic control [43]. In addition, other diseases that increase the risk of *D. folliculorum* infestation or demodicosis, such as kidney damage [44] and dermatological diseases [45] were not evaluated. However, a multivariate analysis was performed to reduce the effect produced by confounding variables described in the literature associated with *D. folliculorum* infestation or demodicosis to whom information was available while the study was conducted. Hence, we consider that the findings of this study can provide a macro view regarding the association between T2D and *D. folliculorum* infestation in the Peruvian population.

## 5. Conclusions

In conclusion, *D. folliculorum* infestation was highly prevalent in people with or without T2D attending a public health center in Palpa, a city in the region of Ica, Peru. Furthermore, the presence of T2D was found to be associated with *D. folliculorum* infestation after adjusting for potential confounding variables. In this sense, it is important to strengthen surveillance to identify *D. folliculorum* infestation or demodicosis in people with T2D, especially in low- and middle-income countries with an increased prevalence of this chronic disease, such as Peru. This surveillance could positively impact the best choice of treatment and orient the strengthening of primary prevention actions to avoid future reinfestations. Future studies could consider some characteristics, such as the time of illness and treatment for diabetes, to characterize and identify predisposing factors among patients with diabetes for the presence of infestation by *D. folliculorum*.

## Figures and Tables

**Figure 1 ijerph-19-13582-f001:**
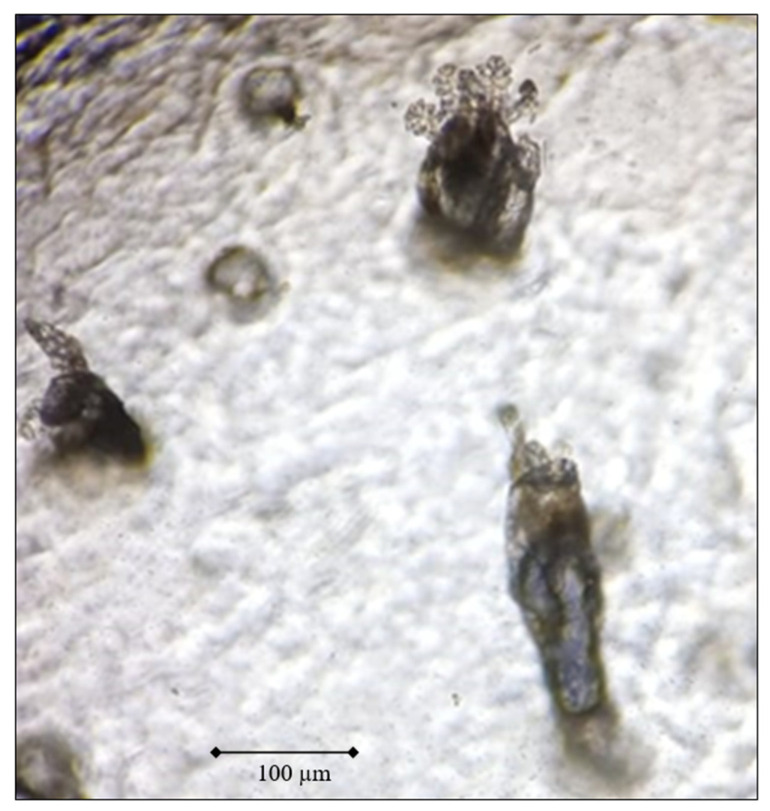
*Demodex folliculorum* obtained by the standardized superficial biopsy technique with cyanoacrylate (Magnification: 40×).

**Table 1 ijerph-19-13582-t001:** Descriptive characteristics of the study sample with or without Type 2 diabetes.

Characteristic	Without T2D (n = 100), n	With T2D (n = 100), n
Sex		
Male	42	41
Female	58	59
^a^ Age (years)	55.0 ± 11.4	57.5 ± 11.2
Domestic animal husbandry		
No	19	25
Yes	81	75
Symptoms associated with demodicosis		
No	99	96
Yes	1	4
Demodex		
No	65	24
Yes	35	76

^a^ Mean ± standard deviation. T2D: Type 2 diabetes.

**Table 2 ijerph-19-13582-t002:** Demodicosis presence according to study sample characteristics.

Characteristic	Demodicosis, n (%)		*p*-Value
	No	Yes	
Sex			
Male	38 (19.0)	45 (22.5)	0.758
Female	51 (25.5)	66 (33.0)	
Age			
<60 years	58 (29.0)	67 (33.5)	0.485
≥60 years	31 (15.5)	44 (22.0)	
Domestic animal husbandry			
No	17 (8.5)	27 (13.5)	0.376
Yes	72 (36.0)	84 (42.0)	
Symptoms associated with demodicosis			
No	89 (44.5)	106 (53.0)	0.043
Yes	0 (0.0)	5 (2.5)	
T2D			
No	65 (32.5)	35 (17.5)	<0.001
Yes	24 (12.0)	76 (38.0)	

T2D: Type 2 diabetes.

## Data Availability

The data presented in this study are available on request from the corresponding author. The data are not publicly available due to privacy considerations.

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
