# Peer review of "Type 2 Diabetes Mellitus and Demodex folliculorum Infestation: A Cross-Sectional Study in Peruvian Patients"

_ijerph, 2022, doi:10.3390/ijerph192013582_

Round 1

Reviewer 1 Report

Dear Authors, I have read your manuscript with interest.

The current manuscript titled: "Type 2 diabetes mellitus and Demodex folliculorum infestation: a cross-sectional study in peruvian patients" represents an important analysis of evolving field of Diabetology and Dermatology.

The title reflects the manuscript content and helps the reader navigate the article essence.

In my opinion, these are the adjustments which should be made to increase the value of your manuscript:

1.       In the manuscript title, please change all the letters to capitals and change “Demodex folliculorum” to italic.

2.       Please do not use personal words (e.g., we enrolled, etc.) in the manuscript and replace them with general ones (e.g., was enrolled, included, etc.).

3.       Methods section:

-        from the text provided, it is not clear how patients were selected for this study, please describe this process in detail;

-        change please „lupus erythematosus” to „systemic lupus erythematosus”;

-        describe in detail the criteria established by American Diabetes Association for T2DM diagnosing;

-        this study presents very few patient characteristics that could explain the presence of this associated pathology in patients with T2DM. The patients’ number is quite small, so add more patients’ characteristics and their comorbidities in both diabetic and non-diabetic subgroups. Such a high percentage of Demodex folliculorum is not clear, perhaps it will become clearer when you explain how patients were included in this study and what other factors contributed to these results.

-        why did the Authors use age distributions of less than and more than 60 years? Try to use a more detailed age distribution. Also, enter the youngest and oldest age;

-        indicate which animals the patients had and what is the statistical distribution for these subgroups, there may be a difference, and, in the conclusions, you can indicate this, for subsequent practical recommendations for patients in everyday practice.

-        also, specify in percentage the patients with T2DM treatment and check the possible connection of these drugs with the Demodex folliculorum presence.

4.       Adapt the bibliography as directed by the journal recommendation.

5.       In Conclusions section, please indicate the practical study significance, as well as the prospects for using this method in real clinical practice.

6.       The manuscript contains some punctuation errors, please revise the text.

Author Response

We thank the reviewers and the editors for their helpful comments and suggestions provided for our manuscript. All comments and reviews have been addressed. Therefore, we are submitting a revised version of the manuscript. We list all comments and changes below, according to the revision process. We tried to address the issues raised as best as possible.

Best regards,

Jaime Rosales

Reviewer 2 Report

The article describes the rarely discussed issue of parasite infestation and its coexistence with chronic diseases. This is an interesting topic, but not new. The authors confirmed what we already know - a number of diseases that affect immunity promote the development of parasitic and fungal diseases. However, the authors' analysis of this particular parasite, which is rarely described in the medical literature, is interesting.

However, there are some ambiguities and a lack of precise data on group recruitment, which makes this work of little value. It can only be seen as a pilot study and a starting point for the development of a larger study from which more practical and general conclusions could be drawn.

There are also many aspects to explain and improve:

1. Currently, the term "diabetes" rather than "diabetes mellitus" is used. Please also use the abbreviation T2D (type 2 diabetes), not T2DM.

2. In the lines68-70 the authors talk about demodecosis being associated with a specific HLA phenotype, suggesting that such a phenomenon is present in T2D patients. Meanwhile, the connection of the HLA system with the formation of diabetes is observed in type 1. This sentence should either be transferred to the previous paragraphs, which generally refer to the functioning of the immune system, or the research methods should be extended to the determination of selected genes of the HLA system in patients with T2D and try to prove their relationship with demodecosis.

3. What criteria were used to determine the "normal" glycemia and HbA1c value (line 93)?

4. Were other immunodeficient or autoimmune diseases included in the exclusion criteria?

5. Was only D.folliculorum species considered in this study? Was the patient diagnosed with D.brevis excluded from the study?

6. Is the approval of the Palpa Hospital Research Office the same as that of the bioethics committee and consistent with the Helsinki Declaration?

7. As far as I understand, the study included patients who had recently been diagnosed with T2D (maximum 2 years before the start of the study) and the analysis itself only proved the presence of the parasite, without any signs of it (except for 5 patients in total - not much). In my opinion, it is difficult to talk about disease, or rather only about contagion. Perhaps it would be much more valuable to observe the group of people with long-term diabetes and compare it with the group with newly diagnosed T2D.

8. I don't feel qualified to judge about the English language and style but please pay attention to the correct use of English terms and their plural forms (e.g. species in singular and plural appearance the same: "species" - lines 34 and 36).

Author Response

(The authors gave the same response as above.)

Round 2

Reviewer 1 Report

Both the first and the revised manuscript version, in my opinion, do not correspond to the level of this Journal.

I do not recommend this article for publication.

Author Response

We thank the reviewers and the editors for their helpful comments and suggestions for our manuscript. All comments and reviews have been addressed. Therefore, we are submitting a revised version of the manuscript. We list all comments and changes below, according to the revision process. We tried to address the issues raised as best as possible.

The authors

Reviewer 2 Report

Thank you very much to the authors for their replies and for making corrections in the text. However, I stand by my decision because of a major problem that the authors did not address:

This is an interesting topic, but not new. The authors confirmed what we already know - a number of diseases that affect immunity promote the development of parasitic and fungal diseases.

However, there are some ambiguities and a lack of precise data on group recruitment, which makes this work of little value. It can only be seen as a pilot study and a starting point for the development of a larger study from which more practical and general conclusions could be drawn.

Author Response

(The authors gave the same response as above.)
